Hypoxic bone marrow mesenchymal stem cell exosomes promote angiogenesis and enhance endometrial injury repair through the miR-424-5p-mediated DLL4/Notch signaling pathway

Xiong Zhenghua 1 2
Hu Yong 1
Jiang Min 3
Liu Beibei 1
Jin Wenjiao 2
Chen Huiqin 4
Yang Linjuan 5
Han Xuesong 1 2 hanxuesong@kmmu.edu.cn
1 Department of Gynecology, First Affiliated Hospital of Kunming Medical University , Kunming, Yunnan , China
2 Department of Gynecology, Yan’an Hospital Affiliated to Kunming Medical University/Yan’an Hospital of Kunming City , Kunming, Yunnan , China
3 Department of Gynecology, Women and Children’s Hospital Affiliated to Qingdao University , Qingdao, Shandong , China
4 Department of Gynecology, Chuxiong Hospital of Traditional Chinese Medicine , Chuxiong, Yunnan , China
5 Department of Gynecology and Obstetrics, Baoshan Hospital of Traditional Chinese Medicine , Baoshan, Yunnan , China
Navaneethabalakrishnan Shobana
Electronic publication date: 2024 Feb 22
Publication date: 2024
Volume: 12
Electronic Location ID: e16953
Received 2023 Aug 7; Accepted 2024 Jan 25
Copyright: © 2024 Xiong et al.
Copyright year: 2024
Copyright holder: Xiong et al.
License: This is an open access article distributed under the terms of the Creative Commons Attribution License, which permits unrestricted use, distribution, reproduction and adaptation in any medium and for any purpose provided that it is properly attributed. For attribution, the original author(s), title, publication source (PeerJ) and either DOI or URL of the article must be cited.
License URL: https://creativecommons.org/licenses/by/4.0/

Keywords: Bone marrow mesenchymal stem cells, Hypoxic exosomes, miR-424-5p, DLL4/Notch, Endometrial injury, Angiogenesis

Funding: National Natural Science Foundation 81960269 Yunnan Province “Ten Thousand People Plan” Famous Doctor Special Talent Project RLHXS20210331 Yunnan Province Medical Leading Talent Project No. L-2019005 This work was supported by the National Natural Science Foundation of China (No. 81960269), the Yunnan Province “Ten Thousand People Plan” Famous Doctor Special Talent Project (No. RLHXS20210331), the Yunnan Province Medical Leading Talent Project (No. L-2019005). The funders had no role in study design, data collection and analysis, decision to publish, or preparation of the manuscript.

==============================
Background

Currently, bone marrow mesenchymal stem cells (BMSCs) have been reported to promote endometrial regeneration in rat models of mechanically injury-induced uterine adhesions (IUAs), but the therapeutic effects and mechanisms of hypoxic BMSC-derived exosomes on IUAs have not been elucidated.

Objective

To investigate the potential mechanism by which the BMSCS-derived exosomal miR-424-5p regulates IUA angiogenesis through the DLL4/Notch signaling pathway under hypoxic conditions and promotes endometrial injury repair.

Methods

The morphology of the exosomes was observed via transmission electron microscopy, and the expression of exosome markers (CD9, CD63, CD81, and HSP70) was detected via flow cytometry and Western blotting. The expression of angiogenesis-related genes (Ang1, Flk1, Vash1, and TSP1) was detected via RT‒qPCR, and the expression of DLL4/Notch signaling pathway-related proteins (DLL4, Notch1, and Notch2) was detected via Western blotting. Cell proliferation was detected by a CCK-8 assay, and angiogenesis was assessed via an angiogenesis assay. The expression of CD3 was detected by immunofluorescence. The endometrial lesions of IUA rats were observed via HE staining, and the expression of CD3 and VEGFA was detected via immunohistochemistry.

Results

Compared with those in exosomes from normoxic conditions, miR-424-5p was more highly expressed in the exosomes from hypoxic BMSCs. Compared with those in normoxic BMSC-derived exosomes, the proliferation and angiogenesis of HUVECs were significantly enhanced after treatment with hypoxic BMSC-derived exosomes, and these effects were weakened after inhibition of miR-424-5p. miR-424-5p can target and negatively regulate the expression of DLL4, promote the expression of the proangiogenic genes Ang1 and Flk1, and inhibit the expression of the antiangiogenic genes Vash1 and TSP1. The effect of miR-424-5p can be reversed by overexpression of DLL4. In IUA rats, treatment with hypoxic BMSC exosomes and the miR-424-5p mimic promoted angiogenesis and improved endometrial damage.

Conclusion

The hypoxic BMSC-derived exosomal miR-424-5p promoted angiogenesis and improved endometrial injury repair by regulating the DLL4/Notch signaling pathway, which provides a new idea for the treatment of IUAs.

Introduction

Intrauterine adhesions (IUAs) refer to endometrial fibrosis and basal layer damage caused by trauma and infection caused by uterine surgery and are a serious threat to women’s health (Leung, Lin & Liu, 2021). Vascular closure of endometrial tissue and fibrosis of the endometrium are the results of IUAs (Lee et al., 2021). At present, hysteroscopic treatment is one of the main methods for treating IUAs, but there are problems such as a high risk of complications during surgery and a high rate of postoperative recurrence (Di Spiezio Sardo et al., 2016). In addition to hysteroscopic treatment, methods such as hormone therapy (Zhang et al., 2017), uterine perfusion (Aghajanova, Cedars & Huddleston, 2018), and hyaluronic acid gel (Wu et al., 2023) have been used to treat IUAs, but these methods are not appropriate for patients with severe IUAs. At present, how to promote the repair of endometrial injury is still a great challenge. Stem cell therapy has been reported to be effective at improving IUA incidence and is a new approach for treating IUAs (Song et al., 2021). The study of stem cells for endometrial repair has attracted much attention. However, the mechanism by which stem cells play a therapeutic role in IUAs remains to be explored, and elucidating the mechanism may provide new ideas for the treatment of IUAs.

Bone marrow mesenchymal stem cells (BMSCs) are a type of stem cell that, as a heterogeneous population distributed in the bone marrow, have the properties of plastic adhesion, immunomodulation, angiogenesis, and tissue repair (Stroncek et al., 2014). BMSCs can be easily isolated and expanded in vitro (Narbona-Carceles et al., 2014). Due to their paracrine and immunomodulatory functions, they can migrate to the site of damaged tissue and differentiate into specific cell types (Fan et al., 2020). Studies have shown that BMSCs exert their therapeutic effects primarily through extracellular vesicles (EVs), while exosomes are the predominant type of EV (Koniusz et al., 2016). Exosomes shuttle mRNAs, miRNAs, and other molecular components to enable cell-to-cell communication and regulate the function of recipient cells (Valadi et al., 2007). For example, Tan, Xia & Ying (2020) reported that miR-29a in BMSC exosomes can inhibit fibrosis during endometrial repair of intrauterine adhesions. Liu et al. (2021) rebated BMSC exosomes and endothelial progenitor cells to improve lipopolysaccharide-induced IUAs caused by acute uterine injury. It has been reported that the migration and survival rate of mesenchymal stem cells cultured under hypoxic conditions are better than those of mesenchymal stem cells cultured under normoxic conditions (Meng et al., 2018). In addition, exosomes produced from hypoxia-pretreated mesenchymal stem cells have been reported to have enhanced therapeutic effects (Kumar & Deep, 2020). Hypoxia (HP) MSC-Exos have been studied for the treatment of spinal cord injury (Liu et al., 2020) and ischemic heart disease (Cheng et al., 2020). However, it is unclear whether exosomes derived from BMSCs under hypoxic conditions contribute to the treatment of IUAs and, if so, whether they are more effective than exosomes derived from BMSCs under normoxic conditions and what mechanisms are involved. Therefore, this study explored the therapeutic mechanism of hypoxic BMSC-derived exosomes in IUAs. This is the first study to investigate the role of hypoxic BMSC-derived exosomes in the treatment of IUAs.

Exosomes reportedly carry different miRNAs, suggesting that exosomes can serve as vectors for miRNAs to transfer and mediate cell-to-cell communication (Stoorvogel, 2012). There is also a large body of research evidence indicating that miRNAs are important participants in the adaptive response of cells to hypoxia, but only a few reports have described the role of hypoxia-related miRNAs in the endometrium; for example, hypoxia-induced miR-20a targeting inhibits the expression of the DUSP2 protein, and the downregulation of DUSP2 expression activates the signaling cascade of extracellular signal-regulated kinase (ERK), thereby promoting angiogenesis and cell proliferation in endometriosis lesions (Lin et al., 2012). In addition, studies have shown that miRNAs are key regulators of endothelial cell function and are particularly important regulators of angiogenesis (Wang & Olson, 2009). Angiogenesis in endometrial tissue affects endometrial repair, so angiogenesis is critical for healing IUAs (Chen, Chang & Yao, 2013). Therefore, exosomal miRNAs, which can affect angiogenesis, may be closely related to the progression of IUAs. Studies have shown that the mesenchymal stem cell-derived exosomal miR-424 can be transferred into HUVECs to promote angiogenesis (Gong et al., 2017). In addition, miR-424-5p can activate VEGFR-2 signaling through E2F7 to promote angiogenesis (Teng et al., 2020). miR-424 is a miRNA carried by mesenchymal stem cell exosomes and can effectively promote angiogenesis; therefore, we speculate that miR-424 may play a role in the treatment of IUAs via BMSC-derived exosomes by mediating angiogenesis. The role of miR-424-5p in IUAs has never been reported before, and this will be an exploration of a new molecular mechanism. In addition, studies have shown that the growth of endothelial tip cells in vascular buds marks the beginning of angiogenesis and that the germination of endothelial tip cells is inhibited by Notch signaling (Hellström et al., 2007). The Notch receptor is activated by binding to its ligand, Delta-like 4 (DLL4), to control a variety of growth and differentiation processes (Benedito et al., 2009). Using starBase (https://rnasysu.com/encori/), we predicted that DLL4 was the target protein of miR-424-5p. miRNAs can regulate gene expression at the posttranscriptional level by binding to specific mRNAs and inducing their degradation and/or translational repression (Krol, Loedige & Filipowicz, 2010). Therefore, based on the targeted regulatory relationship between miR-424-5p and DLL4, we further hypothesize that exosomal miR-424-5p may promote angiogenesis by mediating the DLL4/Notch signaling pathway and thus play a role in the repair of endometrial injury.

From the new perspective of the use of hypoxic BMSC exosomes in the treatment of IUAs, this study proposed a mechanism by which the new molecule miR-424-5p promotes angiogenesis and improves endometrial injury by regulating DLL4. To test this inference, we established cell and animal models to investigate the mechanism of hypoxia BMSC-derived exosomal miR-424-5p in endometrial injury repair. This study may reveal a new treatment method for endometrial injury repair and provide a theoretical basis for future research on endometrial injury repair.

Materials and Methods

Cell culture

Human umbilical vein endothelial cells (HUVECs) were purchased from Otwo Biotench (HTX3606; Shenzhen, China), and BMSCs were purchased from Otwo Biotench (HTX1946; Shenzhen, China). The cells were cultured in high-glucose DMEM supplemented with 10% fetal bovine serum (FBS) and incubated at 37 °C in a 5% CO2 incubator until the cell density reached 80%. A coculture system of HUVECs and exosomes was established using a Transwell system. HUVECs and 100 μg/ml exosomes were cultured in the upper and lower chambers of a Transwell system for 24, 48, 72 and 96 h at 37 °C in a 5% CO2 incubator. To verify that miR-424-5p facilitates angiogenesis through the DLL4/Notch pathway, 500 ng/ml Jagged1, an activator of the Notch pathway, or 10 μmol/L DAPT, an inhibitor of the Notch pathway, was added to the cell culture dish at 37 °C. The cells were cultured in a 5% CO2 incubator for 24 h and then harvested for analysis.

Cell transfection

BMSCs were collected in the logarithmic growth phase, flushed with PBS, digested with 0.25% trypsin solution, seeded in 24-well plates, and then cultured for 24 h in 5% CO2 at 37 °C. The cells were then transfected with Lipofectamine® 3000 reagent (Invitrogen, Waltham, MA, USA), the miR-424-5p inhibitor, the miR-424-5p mimic and the corresponding negative controls (NC-inhibitor, NC-mimic), pc-DNA, and pc-DLL4. The transfection efficiency was measured 48 h after transfection, and the cells were used for subsequent experiments after successful transfection.

Animal model

In this study, female SD rats aged 8 weeks and weighing 180–220 g (bought from the Animal Experimental Center of Kunming Medical University) were selected as the experimental animals and kept in the same cage. This study was authorized by the Animal Ethics Committee of Kunming Medical University (Approval number: kmmu20220879), and all methods conformed to the stipulations of the National Institutes of Health Guidelines for the Care of Laboratory Animals. The rats were adaptively fed for 1 week under standard SPF quality conditions in an environment with a temperature of 22–26 °C, a relative humidity of 52–58%, and a 12 h/12 h light-dark cycle. Thirty-six mice were randomly divided into three groups: the normal control group (Group 1), the IUA group (Group 2), and the IUA+PBS group (Group 3). Group 1 rats (n = 6) were healthy and did not undergo any manipulation. The Group 2 and Group 3 rats were healthy, and an IUA rat model was established by anesthetizing the rats. Then, the skin (approximately 2 cm long) was longitudinally incised approximately 1 cm above the urethral orifice, and the endometrium was scraped with a small ear-picking spoon until the uterine wall became rough and pale. Finally, the uterus was flushed with normal saline, and the wound was sutured. Group 3 (five rats) received intrauterine injections of 0.2 ml of PBS in addition to completing the curettage procedure.

On the 14th day after surgery, Group 2 (25 rats) were randomly divided into five groups with five rats in each group and treated according to the treatment group. Group A (IUAs) rats received no treatment. Group B rats (IUAs+BMSC-hypo-exos) were treated with BMSC-hyp-exos. Group C rats (IUAs+miR-424-5p mimic) were treated with the miR-424-5p mimic. Group D rats (IUAs+miR-424-5p mimic+pc-DLL4) were treated with the miR-424-5p mimic and overexpressing DLL4. Group E rats (IUAs+miR-424-5p mimic+Jagged1) were treated with a miR-424-5p mimic and a Notch pathway activator (Jagged1).

Purification and characterization of exosomes

BMSC-exos were purified by differential centrifugation. BMSCs were cultured in serum-free DMEM for 48 h. The supernatant was then centrifuged using a differential centrifugation method, after which the cells were collected and concentrated. The concentrated supernatant was loaded onto a 30% sucrose/D2O buffer pad (5 ml, density 1.210 G/cm3), followed by ultracentrifugation at 100,000 × g for 3 h. After the test tube was centrifuged, the liquid components at the bottom of the test tube were collected, flushed with PBS, and centrifuged at 1,500 × g and 100 kDa MWCO for 30 min. The supernatant and exosomes were subsequently filtered through a 0.22 μm filter and stored at −70 °C. CD9, CD63, CD81 and HSP70 were identified via western blotting. The morphology of the purified exosomes was observed via transmission electron microscopy (Hitachi H-7650; Hitachi, Tokyo, Japan).

Exosome labeling and internalization

BMSC-exo was traced with PKH67 (HR8659; Beijing Biotech Co., Ltd., Beijing, China) dye. The labeled exosomes were then suspended in a 100-kDa MWCO hollow fiber membrane, and PBS was used as the negative control. HUVECs (5 × 103 per well) were seeded in 96-well plates, hatched with traced exosomes for 4 h at 37 °C, flushed with PBS and immobilized in 4% paraformaldehyde. Finally, a confocal image was obtained with an Array Scan VTI (Thermo Fisher Scientific, Waltham, MA, US).

Identification by flow cytometry

The cells were resuspended into single cells, after which 10 μL of fluorescein isothiocyanate (FITC)-labeled anti-CD9, anti-CD63 and anti-CD81 antibodies (Abcam, Cambridge, UK) were added. The cells were then incubated at 4 °C for 20 min in the dark, flushed twice with PBS, and resuspended in 1 ml of PBS. Flow cytometry was used to measure fluorescence intensity, and CD9, CD63, and CD81 levels were analyzed using Flow Jo 7.2 software.

Anoxic treatment

After counting, BMSCs were seeded into 10 cm dishes at a mass of 1 × 107 after they were cultured in complete substrate supplemented with 10% FBS for 12 h, after which the medium was replaced with 10 ml of serum-free medium per dish (DMEM/F12) after full adherence. The cells were then cultured in anoxic incubators (0.5% O2, 5% CO2) or normoxic incubators (21% O2, 5% CO2) for 48 h.

CCK-8 cell viability assay

Cell activity was measured by CCK-8 according to the manufacturer’s instructions. Cells were seeded into 96-well plates at 5 × 103 cells/well and cultured for the indicated durations. Then, 10 µl of CCK-8 reagent was added to the substrate in each well. After 1 h of incubation, the optical density (OD) values were read by a microplate reader at a wavelength of 450 nm. This process was repeated for three wells at each time point, and the experiment was independently executed three times.

Western blot

The total protein was extracted from the cells and tissues, and the protein concentration was determined using a BCA test kit. The samples were separated with 10% SDS‒PAGE, transferred to PVDF membranes, soaked in 5% skim milk powder, and blocked at room temperature for 2 h. Prediluted primary antibodies against CD9 (1:500), HSP70 (1:300), CD63 (1:300), CD81 (1:300), DLL4 (1:500), Notch1 (1:500), Notch2 (1:2,000), and GAPDH (1:500) were added, and the samples were subsequently incubated overnight at 4 °C. After the membrane was flushed, the samples were incubated with an enzyme-labeled secondary antibody (1:2,000) at room temperature for 2 h. Quantitative tests were performed using an enhanced chemiluminescence (ECL) kit, and the band gray values were analyzed via ImageJ.

Immunofluorescence assay

HUVECs were grown on glass covers overnight to prepare cell slides, immobilized with 4% paraformaldehyde for 20 min, permeabilized with 0.1% Triton X-100 for 15 min, blocked with 5% bovine serum albumin at room temperature for 1 h, flushed with PBS, incubated with a primary antibody against CD34 (Ab81289, 1:50) overnight at 37 °C, incubated with a fluorescein isothiocyanate-labeled secondary antibody for 2 h, and finally incubated with Hoechst 33342 (1:200; Sigma‒Aldrich, St. Louis, MO, US) for 15 min in the dark for counterstaining. The cells were then observed using a fluorescence microscope.

RT–qPCR

Total RNA was extracted using TRIzol reagent and reverse transcribed into cDNA. RT‒qPCR was performed using SYBR Green mix, with U6 and GAPDH serving as internal controls. The detailed RT‒qPCR primer sequences are shown in Table 1, and the relative gene expression was calculated using the 2−ΔΔCt method.

Table 1 Primer sequences for RT-qPCR.

Genes	Sequence (F: Forward primer, R: Reversed primer)	
miR-424-5p	F: 5′-CGAGGGGATACAGCAGCAAT-3′
R: 5′-TTCCCCACGAGGGGGTATAG-3′	
Ang1	F: 5′-TGGTGGTTTGATGCTTGTGG-3′
R: 5′-GTTTCTCACCTGGCAGCTTC-3′	
Flk1	F: 5′-AGAGACCGGCTGAAGCTAGG-3′
R: 5′-GGAAGAGATGGCCTGGTAAACA-3′	
Vash1	F: 5′-GAGGGGTCAAGGTGAGTTCA-3′
R: 5′-GGGTGAGGGGAACATGAGAA-3′	
TSP1	F: 5′-CAGGTAGGCTGTGGAATTGC-3′
R: 5′-GTCACAGTCATCAGGGCAAC-3′	
U6	F:5′-CCCCTGGATCTTATCAGGCTC-3′
R: 5′- GCCATCTCCCCGGACAAAG-3′	
GAPDH	F:5′-GGAGTCCACTGGTGTCTTCA-3′
R: 5′-GGGAACTGAGCAATTGGTGG-3′	

Tissue staining

The rat uteruses were flushed with PBS, and paraffin sections of the tissue were prepared. The sections were routinely dewaxed and dyed with hematoxylin for 1 min for HE staining. The plants were then flushed with tap water, differentiated with 1% hydrochloric acid for 10 s, blued with 1% ammonia water for 5 s, dyed with eosin for 2 min, and sealed after dehydration for observation and analysis. For the immunohistochemical reaction, the sections were routinely dewaxed, incubated with primary antibodies against CD31 (ab28364, 1:50) and VEGFA (ab52917, 1:100) overnight at 4 °C, flushed with PBS, and incubated with secondary antibodies (1:1,000) overnight at 4 °C. After the excess PBS was removed from the sections, the sections were chemically stained, dehydrated, and sealed for observation and analysis.

Angiogenesis test

Matrigel (50 μL) was added to each well, after which the cells were allowed to polymerize for the endothelial tube formation assay. The cells in each group were then seeded on Matrigel in a 96-well plate at a mass of 1 × 105 cells, observed by microscopy, and then photographed after hatching at 37 °C for 12 h.

Target binding site prediction dual-luciferase gene reporter experiment

The wild-type (WT) 3′-UTR of DLL4 cDNA was synthesized and cloned using the starBase database to identify the site at which miR-424-5p binds to DLL4, which was subsequently input into the pMIR luciferase miRNA target expression reporter to generate the WT-DLL4-3′-UTR. Based on the name of the WT-DLL4-3′-UTR, the nucleotide that may bind to miR-424-5p to generate a mutant DLL4-3′-UTR, the resulting vector was named MUT-DLL4-3′-UTR. These vectors were then transiently transfected with the miR-424-5 p mimic or NC mimic into 293T cells using Liposome 3000 reagent. Luciferase activity was then assessed after 48 h of transfection, and the Renilla luciferase vector was used as the internal control.

Statistical analysis

GraphPad Prism 5.0 (GraphPad Software, La Jolla, CA, USA) was used for analyzing the experimental data and for plotting the graphics. Student’s t test and one-way analysis of variance were used for statistical analysis. P < 0.05 was considered to indicate statistical significance.

Results

Hypoxic BMSC-derived exosomes significantly promoted the proliferation and angiogenesis of HUVECs

The effects of normal oxygen BMSC exosomes and hypoxic BMSC exosomes on the proliferation and angiogenesis of HUVECs were observed. First, the extracted exosomes were identified. The morphology of the purified exosomes observed by transmission electron microscopy was characterized by the presence of circular vesicles (Fig. 1A). Flow cytometry was used to detect the exosome markers CD9, CD63 and CD81 (which are involved in the transport of exosomes on the membrane), and the results showed that the cells were all positive for the exosome markers CD9, CD63 and CD81 (Fig. 1B). Western blotting also detected the expression of the exosome markers CD9, CD63, CD81 and HSP70 (Fig. 1C). This finding indicated that we successfully extracted the exosomes. We cocultured exosomes with HUVECs to observe the proliferation and angiogenesis of HUVECs induced by the exosomes. It was first confirmed that HUVECs can take up exosomes by PKH67 fluorescence staining; that is, BMSCs-exos with green fluorescence were observed in the nuclei of blue HUVECs (Fig. 1D). Then, HUVECs that were not cocultured with exosomes were taken as the normal control group (NC), and HUVECs that were cocultured with BMSC exosomes treated with normal oxygen (the BMSCs-norm-exo group) and BMSC exosomes treated with hypoxia (the BMSCS-hyp-exo group) were taken as the experimental group. The proliferation of HUVECs detected by a CCK-8 assay showed that, compared with that in the NC group, the viability of the cells in the BMSC-norm-exo and BMSC-hyp-exo groups was significantly greater, and the ability of the BMSC-hyp-exo group to promote cell proliferation was greater than that of the BMSC-norm-exo group (Fig. 1E). The expression of Ang1 (maintaining vascular stability), Flk1 (encoding vascular endothelial growth Factor A receptor), Vash1 (inhibiting angiogenesis), and TSP1 (inhibiting angiogenesis) was detected by RT‒qPCR. The results showed that the expression levels of Ang1 and Flk1 were significantly upregulated after coculture with exosomes. The expression levels of Vash1 and TSP1 were significantly downregulated, and the regulatory effect of exosomes on angiogenesis-related genes was more significant after hypoxia treatment (Figs. 1F–1I). Vascular formation experiments showed that HUVECs had a denser and more extended tubular network after coculture with exosomes, and the effect of the BMSC-hyp-exo group was greater than that of the BMSC-norm-exo group (Fig. 1J). These results indicate that BMSC-derived exosomes can promote the proliferation and angiogenesis of HUVECs and that the stimulatory effect of hypoxic BMSC-derived exosomes is more significant than that of normally oxygenated BMSC-derived exosomes.

Figure 1 Hypoxic BMSC-derived exosomes significantly promoted the proliferation and angiogenesis of HUVECs.

(A) Exosome morphology was observed by transmission electron microscopy. (B) Flow cytometry was used to detect the exosomal markers CD9, CD63 and CD81. (C) The expression of exosome markers (CD9, CD63, CD81 and HSP70) was detected by Western blot. (D) Exosome uptake was observed by PKH67 staining (scale bar = 100 μm). (E) CCK-8 was used for detecting cell proliferation. (F–I) Expression of Ang1, Flk1, Vash1, and TSP1 was detected by RT‒qPCR. (J) An angiogenesis test was performed to measure angiogenesis ability. Compared with NC group, **P < 0.01, ***P < 0.001.

Hypoxic BMSC-exos promote cell proliferation and angiogenesis through miR-424-5p

Through RT‒qPCR, we observed that miR-424-5p was highly expressed in the exosome group compared with the NC group, and its expression in the BMSC-hyp-exo group was significantly greater than that in the BMSC-norm-exo group (Fig. 2A). These results indicated that hypoxia promoted the expression of the BMSC-derived exosomal miR-424-5p. To further clarify the mechanism by which hypoxia-related BMSC-exos regulate cell proliferation and angiogenesis via miR-424-5p, a miR-424-5p inhibitor and miR-424-5p mimic were transfected into BMSCs, and exosomes were extracted from the BMSCs after hypoxia treatment. The exosomes were then cocultured with HUVECs. First, the transfection efficiency was confirmed. The results showed that the expression of miR-424-5p was significantly downregulated in the co-hyp-exo+miR-424-5p inhibitor group and significantly upregulated in the co-hyp-exo+miR-424-5p mimic group, indicating successful transfection (Fig. 2B). The proliferation of HUVECs was detected by CCK-8, and the results showed that the proliferation of HUVECs in the co-hyp-exo group was greater than that in the NC group. However, proliferation decreased in the co-hyp-exo+miR-424-5p inhibitor group compared with the co-hyp-exo+NC-inhibitor group. Compared with that in the co-hyp-exo+NC-mimic group, the proliferative activity in the co-hyp-exo+miR-424-5p mimic group was increased (Fig. 2C). These results indicated that the ability of exosomes to promote HUVEC proliferation was weakened by transfection of the miR-424-5p inhibitor, while the opposite effect was observed with transfection of the miR-424-5p mimic. The expression of the proangiogenic genes Ang1 and Flk1 and the antiangiogenic genes Vash1 and TSP1 was detected by RT‒qPCR. The results showed that the expression of Ang1 and Flk1 was upregulated and that the expression of Vash1 and TSP1 was downregulated after coculture with exosomes. Transfection of the miR-424-5p inhibitor inhibited the expression of Ang1 and Flk1 and promoted the expression of Vash1 and TSP1, while transfection of the miR-424-5p mimic had the opposite effect (Figs. 2D–2J). Finally, angiogenesis experiments showed that angiogenesis was enhanced in the co-hyp-exo group compared with the control group, and after transfection with the miR-424-5p inhibitor, angiogenesis was weakened compared with that in the co-hyp-exo+NC-inhibitor group. After transfection with the miR-424-5p mimic, angiogenesis was further enhanced compared with that in the co-hyp-exo+NC-mimic group (Fig. 2H). These findings suggested that hypoxic BMSC-exos promote cell proliferation and angiogenesis through miR-424-5p.

Figure 2 Hypoxic BMSC-exos promote cell proliferation and angiogenesis through miR-424-5p.

(A) The expression of miR-424-5p was detected by RT‒qPCR. (B) The transfection efficiency of miR-424-5p was confirmed by RT‒qPCR. (C) CCK‒8 was used for detecting cell proliferation. (D–G) The expression of Ang1, Flk1, Vash1, and TSP1 was detected by RT‒qPCR. (H) An angiogenesis test was performed to measure angiogenesis ability. Compared with the NC group, *P < 0.05, **P < 0.01, ***P < 0.001; compared with the co-hyp-exo+NC-inhibitor group, #P < 0.05, ##P < 0.01, ###P < 0.001; compared with the co-hyp-exo+NC-mimic group, $P < 0.05, $$P < 0.01, $$$P < 0.001.

Hypoxic exosome-mediated angiogenesis is related to the DLL4/Notch signaling pathway

Studies have shown that the DLL4/Notch signaling pathway plays an important role in angiogenesis; therefore, we explored whether hypoxic exosomes affect cell angiogenesis through the DLL4/Notch signaling pathway. First, the expression of DLL4/Notch signaling pathway-related proteins (DLL4, Notch1 and Notch2) after coculture of hypoxic exosomes with HUVECs and overexpression of DLL4 (pc-DLL4) was detected. Western blot analysis of the cells cocultured with hypoxic exosomes and HUVECs revealed that the expression levels of DLL4, Notch1 and Notch2 were downregulated, while the expression levels of DLL4, Notch1 and Notch2 were upregulated after the overexpression of DLL4 (Fig. 3A). These findings suggested that the processing of hypoxic exosomes induces changes in the DLL4/Notch signaling pathway and that DLL4 is the key protein involved. Moreover, the formation of HUVECs was detected. We detected the expression of the marker CD34 in endothelial tip cells by immunofluorescence and flow cytometry. The results showed that the expression of the marker CD34 in endothelial tip cells was upregulated after coculture of hypoxic exosomes with HUVECs but downregulated after overexpression of DLL4 (Figs. 3B and 3C). These results suggest that the promotive effect of hypoxic exosomes on CD34 expression can be reversed by overexpression of DLL4. In addition, RT‒qPCR and angiogenesis experiments showed that after hypoxic exosome treatment, the expression of the proangiogenic genes Ang1 and Flk1 was upregulated, the expression of the antiangiogenic genes Vash1 and TSP1 was downregulated, and angiogenesis was significantly enhanced. However, after overexpression of DLL4, the above phenomenon was reversed (Figs. 3D–3H). These findings suggested that hypoxic exosome-mediated angiogenesis is related to the DLL4/Notch signaling pathway.

Figure 3 Hypoxic exosome-mediated angiogenesis is related to the DLL4/Notch signaling pathway.

(A) The expression levels of DLL4, Notch1 and Notch2 were detected by Western blot. (B) Immunofluorescence was used to detect CD34 levels (scale bar = 100 μm). (C) Flow cytometry was also used to detect CD34 levels. (D–G) RT‒qPCR was used to measure Ang1, Flk1, Vash1, and TSP1 levels. (H) Vascularization ability was detected by a vascularization assay. Compared with the NC group, *P < 0.05, **P < 0.01, ***P < 0.001, ****P < 0.0001; compared with the co-hyp-exo+pc-DNA group, #P < 0.05, ##P < 0.01, ###P < 0.001, ####P < 0.0001.

miR-424-5p targets the expression of DLL4 and affects angiogenesis

Using starBase, we predicted that DLL4 is a miR-424-5p-binding protein (Fig. 4A). A dual-luciferase reporter gene experiment showed that transfection of the miR-424-5p mimic significantly reduced the fluorescence activity of the WT-DLL4 group but had no effect on the fluorescence activity of the MUT-DLL4 group, which further confirmed the targeted binding relationship between miR-424-5p and DLL4 (Fig. 4B). Western blot analysis of the miR-424-5p mimic group revealed that the expression of DLL4 was downregulated compared with that in the NC-mimic group and upregulated in the miR-424-5p inhibitor group compared with that in the NC-inhibitor group (Fig. 4C). These findings suggested that miR-424-5p can target and negatively regulate the expression of DLL4. To further clarify that miR-424-5p affects angiogenesis by regulating the DLL4 protein, we also overexpressed DLL4 (pc-DLL4) after transfecting cells with the miR-424-5p mimic and evaluated the effects of DLL4 on the effects of miR-424-5p. First, Western blot analysis showed that transfection of the miR-424-5p mimic inhibited the expression of DLL4, Notch1 and Notch2. However, after overexpression of DLL4, the effect of the miR-424-5p mimic was reversed to a certain extent, and the expression of DLL4, Notch1, and Notch2 was promoted (Fig. 4D). Immunofluorescence and flow cytometry showed that the miR-424-5p mimic promoted the expression of CD34, and overexpression of DLL4 reversed the effect of the miR-424-5p mimic (Figs. 4E and 4F). The RT‒qPCR results showed that the expression of Ang1 and Flk1 was upregulated and that the expression of Vash1 and TSP1 was downregulated in the miR-424-5p mimic group compared with the NC-mimic group. Compared with those in the miR-424-5p mimic+pc-DNA group, the expression of Ang1 and Flk1 was downregulated, and the expression of Vash1 and TSP1 was upregulated in the miR-424-5p mimic+pc-DLL4 group (Figs. 4G–4J). These results indicated that the effects of the miR-424-5p mimic on the expression of Ang1, Flk1, Vash1 and TSP1 were also reversed by overexpression of DLL4. Finally, angiogenesis experiments also showed that the promoting effect of the miR-424-5p mimic on angiogenesis was reversed by overexpression of DLL4 (Fig. 4K). This finding suggested that the effect of miR-424-5p on HUVEC angiogenesis occurs through the regulation of DLL4.

Figure 4 miR-424-5p targets the expression of DLL4 and affects angiogenesis.

(A) The starBase website was used to predict the binding site of miR-424-5p to DLL4. (B) Dual-luciferase gene reporting experiments confirmed the targeted binding relationship between miR-424-5p and DLL4. (C) The expression of DLL4 was detected by Western blotting. (D) The expression levels of DLL4, Notch1 and Notch2 were detected by Western blot. (E) Immunofluorescence was used to detect CD34 levels (scale bar = 100 μm). (F) Flow cytometry was also used to detect CD34 levels. (G–J) RT‒qPCR was used to measure Ang1, Flk1, Vash1, and TSP1 levels. (K) Vascularization ability was detected by a vascularization assay. Compared with the NC-mimic group, *P < 0.05, ***P < 0.001; compared with the miR-424-5p mimic+pc-DNA group, #P < 0.05, ##P < 0.01, ###P < 0.001.

Hypoxic exosomes promote angiogenesis by mediating the DLL4/Notch signaling pathway through miR-424-5p

To clarify the molecular mechanism through which miR-424-5p promotes angiogenesis through the DLL4/Notch signaling pathway, Jagged1 (a Notch pathway activator) and DAPT (a Notch pathway inhibitor) were added after transfection of the cells with the miR-424-5p mimic. The effect of Jagged1 and DAPT on angiogenesis was assessed via RT‒qPCR and angiogenesis assays. The RT‒qPCR results showed that the expression of Ang1 and Flk1 was upregulated and that the expression of Vash1 and TSP1 was downregulated in the miR-424-5p mimic group compared with the NC-mimic group. Compared with those in the miR-424-5p mimic group, the expression of Ang1 and Flk1 was downregulated, the expression of Vash1 and TSP1 was upregulated in the miR-424-5p mimic+ Jagged1 group, the expression of Ang1 and Flk1 was further upregulated, and the expression of Vash1 and TSP1 was further downregulated in the miR-424-5p mimic+DAPT group (Figs. 5A–5D). Angiogenesis experiments showed that angiogenesis was greater in the miR-424-5p mimic group than in the NC-mimic group. Compared with that in the miR-424-5p mimic group, the angiogenesis ability of the miR-424-5p mimic+Jagged1 group was decreased, and the angiogenesis ability of the miR-424-5p mimic+DAPT group was further strengthened (Fig. 5E). These results indicated that activation of the DLL4/Notch signaling pathway could weaken the effect of the miR-424-5p mimic on promoting blood vessel formation. In general, hypoxic exosomes promote blood vessel formation through the DLL4/Notch signaling pathway mediated by miR-424-5p.

Figure 5 Hypoxic exosomes promote angiogenesis by mediating the DLL4/Notch signaling pathway through miR-424-5p.

(A–D) Expression of Ang1, Flk1, Vash1, and TSP1 was detected by RT‒qPCR. (E) An angiogenesis test was performed to measure angiogenesis ability. Compared with NC-mimic group, *P < 0.05, ***P < 0.001; compared with miR-424-5p group, #P < 0.05, ##P < 0.01, ###P < 0.001.

Hypoxic bone marrow mesenchymal stem cell exosomes improve endometrial injury in IUA rats

To test the influence of BMSC-exos on angiogenesis in vivo, we established an IUA model of uterine adhesions in rats. H&E staining revealed that the endometrial cavity surface of the NC group was covered with regularly arranged columnar epithelium, and abundant endometrial glands were observed. Compared with those in the NC group, the gross uterine morphology in the IUA and IUA + PBS groups was contracted, the endometrial glands were noticeably reduced, and tissue hyperplasia was observed. Pathology improved, and the number of endometrial glands increased in the IUA + BMSC-hyp-exo group (Fig. 6A). Immunohistochemical results showed that VEGFA and CD31 levels were lower in the IUA and IUA + PBS groups than in the NC group. VEGFA and CD31 levels were elevated in the IUA + BMSC-hyp-exo group compared to those in the IUA + PBS group (Fig. 6B). These results suggested that treatment with hypoxic BMSC exosomes promoted the expression of angiogenesis-related proteins. RT‒qPCR and Western blotting revealed that miR-424-5p was expressed at lower levels and that DLL4, Notch1, and Notch2 were expressed at higher levels in the IUA and IUA + PBS groups than in the NC group. miR-424-5p was upregulated and DLL4, Notch1, and Notch2 were downregulated in the IUA + BMSC-hyp-exo group compared with those in the IUA + PBS group (Figs. 6C and 6D). In animal experiments, the therapeutic effect of hypoxic BMSC-derived exosomes on IUAs was confirmed to be related to miR-424-5p and the DLL4/Notch pathway. These results indicate that BMSC-derived exosomes facilitate IUA vascularization and ameliorate endometrial damage in vivo.

Figure 6 Hypoxic bone marrow mesenchymal stem cell exosomes improve endometrial injury in IUA rats.

(A) H & E staining was used to observe the uterine sections (scale bar = 250 μm). (B) Immunohistochemistry was used to detect CD31 and VEGFA levels (scale bar = 100 μm). (C) RT‒qPCR was used to measure the level of miR-424-5p. (D) The levels of DLL4, Notch1 and Notch2 were detected by Western blot. Compared with the NC group, **P < 0.01, ***P < 0.001; compared with the IUA+PBS group, #P < 0.05, ##P < 0.01, ###P < 0.001.

miR-424-5p improves endometrial injury in IUA rats by mediating the DLL4/Notch signaling pathway

The mechanism by which miR-424-5p improves IUAs by mediating the DLL4/Notch signaling pathway to promote angiogenesis was further verified in vivo. H&E staining revealed that uterine tissue hyperplasia and endometrial gland density were significantly lower in the IUA group than in the NC group. Compared with that in the IUA group, the pathological status of rats in the IUA+miR-424-5p mimic group was improved. Tissue hyperplasia and endometrial glands from the IUA+miR-424-5p mimic+pc-DLL4 group and IUA+miR-424-5p mimic+Jagged1 group appeared again and were significantly reduced compared with those from the IUA+miR-424-5p mimic group (Fig. 7A). Treatment with the miR-424-5p mimic effectively alleviated endometrial injury in IUA rats, but treatment with the DLL4/Notch pathway activator reversed this effect. Immunohistochemical results showed that VEGFA and CD31 levels were significantly lower in the IUA group than in the NC group. Compared with those in the IUA group, the VEGFA and CD31 levels were greater in the IUA + miR-424-5p mimic group. VEGFA and CD31 levels in the IUA + miR-424-5p mimic + pc-DLL4 group and the IUA + miR-424-5p mimic + Jagged1 group were lower than those in the IUA + miR-424-5p mimic group (Figs. 7B and 7C). The RT‒qPCR and Western blot results showed that, compared with that in the NC group, the expression of miR-424-5p was downregulated, while the expression of DLL4, Notch1 and Notch2 was upregulated in the IUA group. Treatment with the miR-424-5p mimic promoted the expression of miR-424-5p and inhibited the expression of DLL4, Notch1, and Notch2, while pc-DLL4 and Jagged1 reversed the effects of the miR-424-5p mimic (Figs. 7C and 7D). These results suggest that miR-424-5p facilitates IUA vascularization in vivo by mediating DLL4/Notch signaling, ameliorating endometrial injury.

Figure 7 miR-424-5p improves endometrial injury in IUA rats by mediating the DLL4/Notch signaling pathway.

(A) H & E staining was used to observe the uterine sections (scale bar = 250 μm). (B) Immunohistochemistry was used to detect CD31 and VEGFA levels (scale bar = 100 μm) (C) RT‒qPCR was used to measure the level of miR-424-5p. (D) The levels of DLL4, Notch1 and Notch2 were detected by Western blot. Compared with the NC group, ***P < 0.001; compared with the IUA group, ##P < 0.01, ###P < 0.001; compared with the IUA+ miR-424-5p mimic group, $$P < 0.01, $$$P < 0.001.

Discussion

The difficulty of endometrial repair caused by impaired endometrial epithelial cell regeneration and angiogenesis is an important factor in IUA formation (Xu et al., 2017). Studies have shown that endometrial mesenchymal stem cells (MenSCs) (Zhang et al., 2016) or bone marrow mesenchymal stem cells (BMSCs) (Yu et al., 2018) can promote angiogenesis and effectively treat IUAs. We have launched a new exploration of the mechanism of angiogenesis repair in IUAs via the paracrine effects of hypoxia-treated BMSCs and exosomes.

Mesenchymal stem cells (MSCs) are ideal candidates for tissue repair, and studies have shown that MSCs can play a therapeutic role by promoting angiogenesis (Kong et al., 2013; Matsuda et al., 2013). BMSC implantation has been proven to be an effective strategy for the treatment of various diseases, and it is also involved in the regulation of IUA progression (Yao et al., 2019). It is generally believed that transplanted BMSCs usually perform their biological functions by secreting exosomes (Li et al., 2020a, 2020b). Studies have shown that the oxygen concentration is an important factor affecting the proliferation and differentiation of MSCs (Hu et al., 2014), and most BMSCs survive in vivo under hypoxic conditions (Mohyeldin, Garzón-Muvdi & Quiñones-Hinojosa, 2010). In the present study, the exosomes of normal-oxygen BMSCs and hypoxic BMSCs were cocultured with HUVECs, and coculture with exosomes effectively promoted the proliferation and angiogenesis of HUVECs; moreover, the stimulatory effect of hypoxic exosomes was more significant than that of normal-oxygen BMSCs. Our study is consistent with a study by Zhu et al. (2022) who also showed that exosomes from BMSCs under hypoxic conditions had a greater therapeutic effect on ulcerative colitis injury than exosomes under normoxic conditions. Our study also revealed that treatment with hypoxic BMSC-derived exosomes effectively improved endometrial lesions in IUA rats and promoted the expression of VEGFA and CD31. CD34 is the most sensitive marker of the endothelium (Sidney et al., 2014). VEGF is an important proangiogenic factor (Claesson-Welsh & Welsh, 2013). These findings suggested that hypoxic BMSC-derived exosomes can effectively treat IUAs by promoting angiogenesis.

Studies have shown that miRNAs are involved in regulating various aspects of angiogenesis, including endothelial cell proliferation, migration and morphological changes (Liang et al., 2016; Welten et al., 2016). It has also been reported that miRNAs carried by exosomes mediate intercellular signal transduction and significantly affect the biological function of cells (Cheng et al., 2014). For example, miR-424-5p, a miRNA carried in BMSC exosomes, has been shown to attenuate osteogenic development through WIF1/Wnt/β-catenin (Wei et al., 2021). Exosomes derived from BMSCs also inhibit apoptosis and epithelial mesenchymal transformation through miR-424-5p, thereby alleviating diabetic nephropathy (Cui et al., 2022). In conclusion, miR-424-5p is an important miRNA that plays a role in BMSC exosomes. Our study revealed that the expression of miR-424-5p in hypoxic BMSC exosomes was significantly greater than that in normal oxygen BMSC exosomes. Moreover, the ability of hypoxic BMSC exosomes cocultured with HUVECs to promote cell proliferation and angiogenesis was inhibited after transfection with the miR-424-5p inhibitor, while transfection with the miR-424-5p mimic further enhanced the effect of the hypoxic BMSC exosomes. In addition, in the present study, treatment with the miR-424-5p mimic effectively improved endometrial injury in IUA rats and promoted the expression of CD31 and VEGFA. We found a new therapeutic effect of miR-424-5p in IUAs. Hypoxic BMSC-exos alleviated IUA progression by promoting cell proliferation and angiogenesis through miR-424-5p.

Studies have reported that DLL4 can be coupled with Notch signaling to promote angiogenesis and arteriformation (Pitulescu et al., 2017). The regulation of endothelial cells by the DLL4/Notch signaling pathway initiates angiogenesis (Karamysheva, 2008). Other studies have shown that the expression of Notch1 is affected by miR-424-5p and that the overexpression of miR-424-5p can inhibit the expression of Notch1 and Notch2 (Zhou et al., 2017). In this study, starBase was used to predict that DLL4 is the target protein of miR-424-5p, that miR-424-5p can target and negatively regulate the expression of DLL4, and that the miR-424-5p mimic also inhibited the expression of the Notch1 and Notch2 proteins in the Notch signaling pathway. In addition, the stimulating effect of the miR-424-5p mimic on HUVEC angiogenesis or the alleviation of IUA progression could be reversed by the overexpression of DLL4. These results suggest that miR-424-5p promotes angiogenesis by mediating the DLL4/Notch signaling pathway. Studies have shown that DLL4/Notch signaling can regulate the polarization of M2 macrophages (Pagie, Gérard & Charreau, 2018). Moreover, M2 macrophage polarization is closely related to fibrosis, An et al. (2023) and Jiao et al. (2021). Given that endometrial fibrosis is an important feature of IUAs, we hypothesize that miR-424-5p may also inhibit fibrosis in IUAs by mediating the DLL4/Notch signaling pathway, which is expected to constitute a new idea for our subsequent research.

Conclusion

In summary, our study explored the therapeutic effect of hypoxic BMSC-derived exosomes on IUAs and revealed that hypoxic exosomes promote angiogenesis and alleviate IUA progression through the miR-424-5p-mediated DLL4/Notch signaling pathway. These findings have enriched the theoretical basis of stem cell therapy for IUAs, and the proposal of this new mechanism also provides a potential therapeutic target for IUAs. This study has several limitations. First, the sample size of our study was not very large, and the sample size needs to be increased to improve the repeatability of the experiment. Second, the molecular mechanisms proposed in cell experiments have not been fully verified in animal experiments.

Supplemental Information

Supplemental Information 1 Author Checklist-Full.

Supplemental Information 2 Full length uncropped gels/blots.

Additional Information and Declarations

Competing Interests

Author Contributions

Animal Ethics

Data Availability

The authors declare that they have no competing interests.

Zhenghua Xiong conceived and designed the experiments, performed the experiments, analyzed the data, prepared figures and/or tables, and approved the final draft.

Yong Hu conceived and designed the experiments, performed the experiments, analyzed the data, prepared figures and/or tables, authored or reviewed drafts of the article, and approved the final draft.

Min Jiang conceived and designed the experiments, analyzed the data, authored or reviewed drafts of the article, and approved the final draft.

Beibei Liu conceived and designed the experiments, performed the experiments, prepared figures and/or tables, and approved the final draft.

Wenjiao Jin performed the experiments, analyzed the data, prepared figures and/or tables, and approved the final draft.

Huiqin Chen analyzed the data, authored or reviewed drafts of the article, and approved the final draft.

Linjuan Yang performed the experiments, prepared figures and/or tables, and approved the final draft.

Xuesong Han conceived and designed the experiments, analyzed the data, authored or reviewed drafts of the article, and approved the final draft.

The following information was supplied relating to ethical approvals (i.e., approving body and any reference numbers):

Kunming Medical University Ethical Review Committee for Animal Experiments provided full approval for this research (MIN.kmmu20220879).

The following information was supplied regarding data availability:

The data is available at figshare: Xiong, Zhenghua; Hu, Yong; Jiang, Min; Liu, Beibei; Jin, Wenjiao; Chen, Huiqin; et al. (2023). Hypoxic bone marrow mesenchymal stem cell exosomes promote angiogenesis and enhance endometrial injury repair through the miR-424-5p-mediated DLL4/Notch signaling pathway. figshare. Figure. https://doi.org/10.6084/m9.figshare.23292740.v5.

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
