# Peer review of "Hypoxic bone marrow mesenchymal stem cell exosomes promote angiogenesis and enhance endometrial injury repair through the miR-424-5p-mediated DLL4/Notch signaling pathway"

_PeerJ, doi:10.7717/peerj.16953_

## Round 0.1 · original submission · Major Revisions

The manuscript has been assessed by two independent reviewers and I strongly suggest addressing the concerns raised by the reviewers before your paper could be considered for publication.

Extensive revision of the manuscript is highly recommended for better understanding of the manuscript. Figures need to be improved in terms of resolution, labeling, and description. Rationale behind the study needs to be presented more clearly. The manuscript seems to lack sufficient information to validate the significance of the study. The rationale behind using various markers/experiments needs to be explained well.

Reviewer 1 ·

Basic reporting

In the manuscript entitled “Hypoxic bone marrow mesenchymal stem cell exosomes promote angiogenesis and enhance endometrial injury repair through the miR-424-5p-mediated DLL4/Notch signaling pathway”, the authors showed that exosomes containing miR-424-5p secreted from hypoxic bone marrow mesenchymal stem cells enhance proliferation and differentiation of endothelial progenitor cells, subsequently improve intrauterine adhesion (IUA) recovery. And they also found the recovery effect of miR-424-5p is achieved by regulating DLL4/Notch signaling pathway. However, there are some issues should be addressed:

1. When markers like Ang1, Flk1, Vash1, TSP1, CD34, and CD31 are first mentioned, it would be beneficial if the authors could briefly outline their functions.
2. What is “NC-inhibitor” mentioned in the manuscript?
3. Some studies also showed that DLL4/Notch signaling can regulate M2 macrophage polarization. And M2 macrophage polarization is closely related to the fibrosis. ( Front Immunol. 2021 Aug 26;12:735014) (Cells. 2021 Nov 6;10(11):3057) (Br J Pharmacol. 2023 Apr 19. doi: 10.1111/bph.16096.) Is it possible that miR-424-5p can inhibit fibrosis in IUA? It would be better if the authors can discuss it.
4. The unit of scale bar in Figure 7 should be “um” not “px”
5. It would be better if the authors can do quantification for the staining in Figure 7 and 8.

Experimental design

no comment

Validity of the findings

no comment

Reviewer 2 ·

Basic reporting

.

Experimental design

.

Validity of the findings

.

Additional comments

1 The article is not innovative enough for publication in your journal.
2 The research ideas of this article are very confusing.
3 The expression of the article makes it very difficult for readers to understand.
4 The pictures provided in the article are of very low quality.

---

## Round 0.2 · Minor Revisions

The authors have addressed the concerns raised by the reviewers adequately and have improved the quality of the manuscript significantly. However, few minor concerns need to be addressed before being considered for publication.

1. The manuscript needs overall revision for better understanding. Authors are suggested to include more information on the existing knowledge on the topic, gaps in the available literature, rationale, and limitations of the current study.

2.More emphasis is needed on the novelty or significance of the current study.

3. Authors are suggested to provide high resolution figures for the publication.

Reviewer 1 ·

Basic reporting

The authors addressed my concerns.

Experimental design

N/A

Validity of the findings

N/A

Reviewer 2 ·

Basic reporting

See below

Experimental design

See below

Validity of the findings

See below

Additional comments

1. The related work section needs to be organized in a more comprehensive way.
2. The quality of all figures should be improved especially Figure 1A, Figure 1B, Figure 3C, Figure 4F.
3. The author should revise the paper sentence by sentence to improve the expression of this manuscript.
4. I suggest the author add some reasons to investigate miR-424-5p, and it is better to add some results involved data analysis or bioinformatics analysis.
5. The novelty of this paper is vague.
6. Please mention the limitations of your research.
7. The authors are advised to perform more analysis of the experimental results in the 'Result' section.

---

## Round 0.3 · accepted · Accept

The authors have diligently addressed the suggested revisions, have significantly improved the quality of the manuscript and I recommend it for publication.

Reviewer 1 ·

Basic reporting

No comment

Experimental design

No comment

Validity of the findings

No comment

Reviewer 2 ·

Basic reporting

Authors have made complete revisions according to my suggestions. I accept this paper for publication in this journal.

Experimental design

Authors have made complete revisions according to my suggestions. I accept this paper for publication in this journal.

Validity of the findings

Authors have made complete revisions according to my suggestions. I accept this paper for publication in this journal.

Additional comments

Authors have made complete revisions according to my suggestions. I accept this paper for publication in this journal.